# OccFusion: Rendering Occluded Humans with Generative Diffusion Priors

**Adam Sun**[*], **Tiange Xiang**[*], **Scott Delp**, **Li Fei-Fei**[†], **Ehsan Adeli**[†]

Stanford University

{adsun, xtiange}@stanford.edu

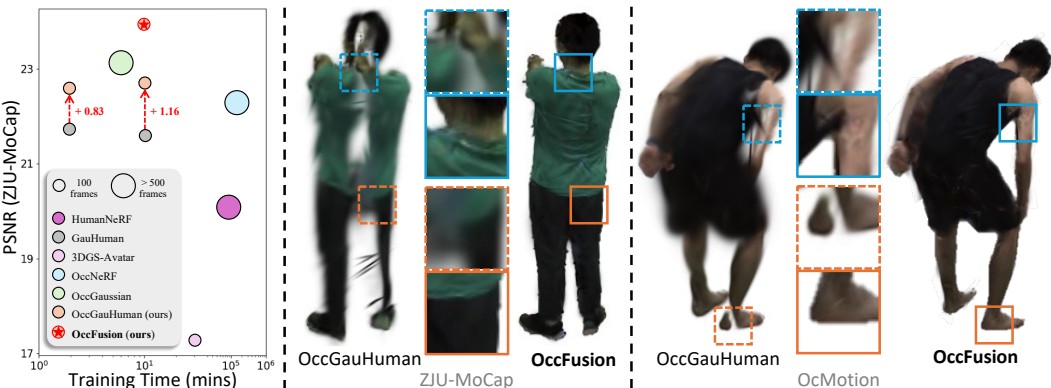

Figure 1: Reconstructing humans from monocular videos frequently fails under occlusion. In this paper, we introduce **OccFusion**, a method that combines 3D Gaussian splatting with 2D diffusion priors for modeling occluded humans. Our method outperforms the state-of-the-art in rendering quality and efficiency, resulting in clean and complete renderings free of artifacts.

## Abstract

Most existing human rendering methods require every part of the human to be fully visible throughout the input video. However, this assumption does not hold in real-life settings where obstructions are common, resulting in only partial visibility of the human. Considering this, we present OccFusion, an approach that utilizes efficient 3D Gaussian splatting supervised by pretrained 2D diffusion models for efficient and high-fidelity human rendering. We propose a pipeline consisting of three stages. In the Initialization stage, complete human masks are generated from partial visibility masks. In the Optimization stage, human 3D Gaussians are optimized with additional supervision by Score-Distillation Sampling (SDS) to create a complete geometry of the human. Finally, in the Refinement stage, in-context inpainting is designed to further improve rendering quality on the less observed human body parts. We evaluate OccFusion on ZJU-MoCap and challenging OcMotion sequences and find that it achieves state-of-the-art performance in the rendering of occluded humans. Project page: `https://cs.stanford.edu/~xtiange/projects/occfusion/`.

## 1   Introduction

Rendering 3D humans from monocular in-the-wild videos has been a persistent challenge, with significant implications in virtual/augmented reality, healthcare, and sports. Given a video of a human

---

[*]Equal contribution; junior author listed first.

[†]Equal mentorship.

38th Conference on Neural Information Processing Systems (NeurIPS 2024).

moving around a scene, this task involves reconstructing the appearance and geometry of the human, allowing for the rendering of the human from novel views.

When faced with the problem of human reconstruction from monocular video, several works based on neural radiance fields (NeRFs) have achieved promising results [37, 57, 19, 9]. 3D Gaussian splatting [24] further improves upon NeRF-based rendering methods for better performance. By representing the human not as an implicit radiance field but as a set of explicit 3D Gaussians, methods like GauHuman [14] and 3DGS-Avatar [46] are able to render humans comparable in quality to NeRF methods, while taking only a few minutes to train and less than a second to render.

While most human rendering studies assume clean, unobstructed environments, real-world settings like hospitals, stadiums, and construction sites involve frequent occlusions. Current methods struggle in these conditions, often producing artifacts, floating elements, or incomplete body parts. Solutions like OccNeRF [64] and Wild2Avatar [63] attempt to address occlusions but are limited by high computational demands and lengthy training times, making them impractical and restricting their real-world applicability.

In this work, we introduce OccFusion, an efficient yet high quality method for rendering occluded humans. To gain improved training and rendering speed, OccFusion represents the human as a set of 3D Gaussians. Like almost all other human rendering methods [64, 63, 67], OccFusion assumes accurate priors such as human segmentation masks and poses are provided for each frame, which can be obtained with state-of-the-art off-the-shelf estimators such as SAM [25] and HMR 2.0 [8]. However, to ensure complete and high-quality renderings under occlusion, OccFusion proposes to utilize generative diffusion priors, more specifically pose-conditioned Stable Diffusion 1.5 [48] with ControlNet [76] plugins, to aid in the reconstruction process.

Our approach consists of three stages: **(1)** *The Initialization Stage:* we utilize segmentation and pose priors to inpaint occluded human visibility masks into complete human occupancy masks to supervise later stages. **(2)** *The Optimization Stage:* we initialize a set of 3D Gaussians and optimize them based on observed regions of the human, applying pose-conditioned Score-Distillation Sampling (SDS) to help ensure completeness of the modeled human body in both the posed and canonical space. **(3)** *The Refinement Stage:* we utilize pretrained generative models to inpaint unobserved regions of the human with context from partial observations and renderings from the previous stage, further improving the quality of the renderings. Despite taking only 10 minutes to train, our method outperforms the state-of-the-art in rendering humans from occluded videos.

In summary, our contributions are: *(i)* We propose OccFusion, the first method to combine Gaussian splatting with diffusion priors for the rendering of occluded humans from monocular videos. Multiple novel components are proposed along with a three-stage pipeline consisting of Initialization, Optimization, and Refinement stages. *(ii)* We demonstrate that OccFusion achieves state-of-the-art efficiency and rendering quality of occluded humans on both simulated and real-world occlusions.

## 2 Related Work

### 2.1 Neural Human Rendering

Traditional methods to reconstruct humans usually require dense arrays of cameras [10, 4, 2] or depth information [71, 50, 4, 5], both of which are unobtainable for in-the-wild scenes. To solve this problem, Neural Radiance Fields (NeRFs) [37] have recently been used to model dynamic humans from monocular videos [57, 9, 19, 17, 72, 52]. These methods achieve high-quality novel view synthesis by parametrizing the human body using an SMPL [35] pose prior and modeling it as a radiance field. However, since NeRFs depend on large Multi-Layer Perceptrons (MLPs), they are computationally expensive, usually taking days to train and minutes to render [24, 14, 46]. To speed up NeRF-based models, multi-resolution hash encoding [40, 43, 6, 18], and generalizability [41, 2, 13, 27] have been proposed. However, these methods either face a rendering bottleneck [14] or an expensive pre-training process, both of which affect their efficiency.

Point-based rendering methods like 3D Gaussian splatting [24] greatly accelerate the rendering of static and dynamic scenes. Recently, there have been an abundance of works applying 3D Gaussian splatting to human rendering tasks [46, 14, 26, 38, 74, 33, 68, 20, 29, 28, 42, 12]. Like NeRF-based approaches, Gaussian splatting-based approaches represent the human in a canonical space

and use Linear Blend Skinning (LBS) to transform the human into the posed space. Gaussian splatting methods achieve state-of-the-art performance of dynamic humans with fast training times and real-time rendering, causing them to be the more desired method [46, 14].

### 2.2 Occluded Human Rendering

Reconstructing complex scenes in the wild is a well-studied problem. NeRF-W [36] and other works [3, 47, 75] are able to account for photometric variation and transient occluders, allowing them to render consistent representations from unconstrained image collections. However, these works are not designed to handle dynamic objects like humans.

Rendering humans in occluded settings, on the other hand, is relatively understudied. Sun *et al.* [51] utilize a layer-wise scene decoupling model to decouple humans from occluding objects. OccNeRF [64] combines geometric and visibility priors with surface-based rendering to train a human NeRF model, while Wild2Avatar [63] proposes an occlusion-aware scene parametrization scheme to decouple the human from the background and occlusions. While these works provide decent renderings of humans free of occlusions, they are slow and impractical due to their usage of NeRFs. A concurrent work to ours is OccGaussian [67], which also proposes to model occluded humans with 3D Gaussians by performing an occlusion feature query in occluded regions. We provide comparisons to their published results in Table 1.

### 2.3 Generative Diffusion Priors

Inferring the appearance of unobserved regions of 3D scenes requires the usage of generative models. The recent success of 2D diffusion models has made them the preferred model to use for generation [54, 21, 34, 48, 32]. To lift 2D diffusion models for 3D content generation, DreamFusion [45] proposed Score Distillation Sampling (SDS), a commonly used method for utilizing a pre-trained 2D diffusion model to supervise 3D content generation [30, 70, 53, 55].

Diffusion models can also be used as priors for training NeRFs and Gaussian splatting, combining reconstruction with generation [61, 78, 79, 62, 66, 73]. ReconFusion [61] uses SDS in conjunction with multi-view conditioning to synthesize the appearance of unobserved regions of a scene from sparse views, while BAGS [78] utilizes SDS to supervise a Gaussian splatting model.

## 3 Preliminaries

Before introducing our method, we provide an overview of key fundamentals in 3D human modeling using SMPL (subsection 3.1). Then, we discuss 3D Gaussian splatting, and how it can be applied to human modeling (subsection 3.2). Finally, we propose OccGauHuman, a simple improvement of GauHuman [14] that is better designed for occluded human rendering (subsection 3.3).

### 3.1 3D Human Modeling

SMPL [35] is a model that parametrizes the human body with a 3D surface mesh. To transform between the canonical space to a pose space, the Linear Blend Skinning (LBS) algorithm is used. Given a 3D point $\mathbf{x_c}$ in the canonical space and the shape $\beta$ and pose $\theta$ parameters of the human, a point in the posed space can be calculated as:

$$\mathbf{x_P} = \sum_{k=1}^{K} w_k \left( G_k(\mathbf{J}, \theta)\mathbf{x_c} + b_k(\mathbf{J}, \theta, \beta) \right), \tag{1}$$

where $J$ contains $K$ joint locations, $G_k$ and $b_k$ are the transformation matrix and translation vector, and $w_k \in [0, 1]$ are a set of skinning weights. The SMPL representation is commonly used as a geometric prior for human rendering [64, 63, 46, 14, 57, 19, 72].

### 3.2 Human Rendering with 3D Gaussian Splatting

**3D Gaussian splatting.** 3D Gaussian splatting [24] models a scene as a set of 3D Gaussians $\Pi$. Each Gaussian is defined by its 3D location $\mathbf{p_i}$, opacity $o_i \in [0, 1]$, center $\mu_\mathbf{i}$, covariance matrix $\Sigma_i$, and

spherical harmonic coefficients. The $i$-th Gaussian is defined as $o_i e^{-\frac{1}{2}(\mathbf{p}-\mu_i)^T \Sigma_i^{-1}(\mathbf{p}-\mu_i)}$. During rendering, these 3D Gaussians are mapped from the 3D world space and projected to the 2D image space via $\alpha$-blending, with the color of each pixel being calculated across the $N$ 3D Gaussians as:

$$C = \sum_{j=1}^{N} c_j \alpha_j \prod_{k=1}^{j-1} (1 - \alpha_k), \tag{2}$$

where $c_j$ is the color and $\alpha$ is the $z$-depth ordered opacity. During the training process, 3D Gaussians are adaptively controlled via densification (splitting and cloning) and pruning until they achieve the optimal density to adequately represent the scene.

**GauHuman [14]** . In the line of work that uses 3D Gaussian splatting for human rendering [46, 26, 29, 12], GauHuman is a representative approach due to its balance between efficiency and rendering quality. After initializing 3D Gaussians on the vertices of the SMPL mesh, GauHuman learns a representation of the human in canonical space and utilizes LBS to transform each individual Gaussian into the posed space. A pose refinement module $MLP_{\Phi_{pose}}$ and an LBS weight field module $MLP_{\Phi_{lbs}}$ are used to learn the LBS transformation, and a merge operation based on KL divergence is used along with splitting, cloning, and pruning to help the 3D Gaussians reach convergence.

We base our method on GauHuman due to its fast training and state-of-the-art representative ability. GauHuman's code is distributed under the S-Lab license and can be accessed here.

### 3.3 OccGauHuman: An Improved Baseline for Occlusion Handling

In common human rendering tasks, videos are captured in a clean environment, with every pixel in the image belonging to either the human or the background. By using a semantic segmentation model such as SAM [25] to preprocess a video, we can train the human rendering model only on pixels labeled as "human". However, occlusions in the videos may lead to sparse observations of the human. As a result, fitting NeRF-based human rendering models on only the visible human pixels results in an incomplete geometry with lots of artifacts [64, 63].

Gaussian splatting-based rendering models [24] are especially suitable for human modeling tasks due to their explicit geometry and point-based representation. In this section, we present three straightforward tweaks of GauHuman [14] to make it perform better on videos with occlusions: *(1)* Firstly, as discussed above, we train the model on visible human pixels only, ensuring that occlusions do not result in learned sparsity on the human model. *(2)* We adjust the loss weights to put more weight on the mask loss computed between rendered human occupancy maps and the segmentation masks — we found that this helps learn more crisp human boundaries. *(3)* We disable the densification and pruning of 3D Gaussians during training — this helps maintain a rather complete human geometry based on the SMPL initialization.

The resulting OccGauHuman model serves as an improved baseline for occluded human reconstruction and as a starting point for our method. Benefits brought by our updates compared to the original GauHuman are presented in Table 1, as well as in Figure 7.

## 4   OccFusion

In our approach, we train a Gaussian splatting-based human rendering model on the visible pixels of a human. However, recovering occluded content for a dynamically moving human is not trivial — humans are usually in challenging poses, and complex occlusions can cause additional issues. It is also essential to preserve a consistent human appearance and geometry across different frames. Considering these challenges, we propose our method OccFusion in multiple separate stages. In the Initialization stage (section 4.1), we inpaint occluded binary human masks for more reliable geometric guidance. In the Optimization stage (section 4.2), we use the inpainted masks to train a human rendering model based on GauHuman [14] while using Score Distillation Sampling (SDS) constraints on both the posed space and canonical space. In the Refinement stage (section 4.3), we fine-tune the trained model from the Optimization Stage with in-context inpainting to further refine the appearance of the human. An overview of our OccFusion is shown in Figure 2.

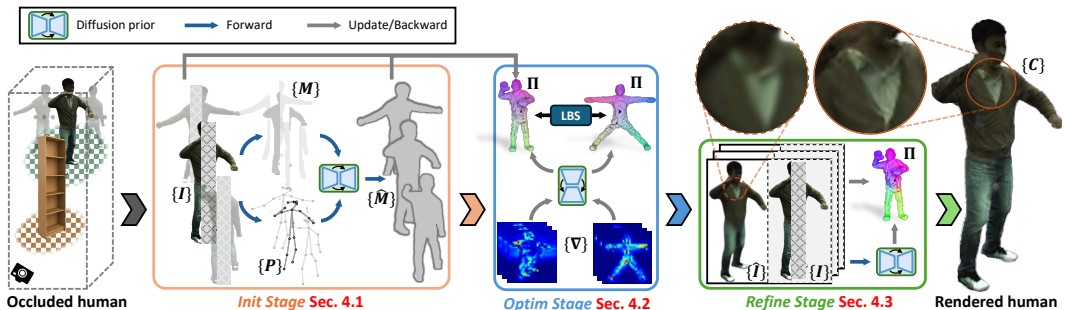

Figure 2: **OccFusion** achieves occluded human rendering via three sequential stages. In the **Initialization Stage**, we recover complete binary human masks $\{\hat{\mathbf{M}}\}$ from occluded partial observations $\{\mathbf{I}\}$ with the help of segmentation priors $\{\mathbf{M}\}$ and pose priors $\{\mathbf{P}\}$. $\{\hat{\mathbf{M}}\}$ will be further used to help optimize the 3D Gaussians $\Pi$ in subsequent stages. In the **Optimization Stage**, we apply $\{\mathbf{P}\}$ conditioned SDS on both posed human and canonical human to enforce the human occupancy to remain complete. In the **Refinement Stage**, we use the coarse human renderings $\{\hat{\mathbf{I}}\}$ from the Optimization Stage to help generate missing RGB values in $\{\mathbf{I}\}$ through our proposed in-context inpainting. Through this process, both the appearance and geometry of the human are fine-tuned to be in high fidelity. Training of all three stages takes only **10 minutes** on a single Titan RTX GPU.

## 4.1 Initialization Stage: Recovering Human Geometry from Partial Observations

Generative diffusion models [48] have demonstrated promise to be used as priors for different tasks [22, 53]. The most straightforward method is to utilize a precomputed segmentation prior $\mathbf{M}$ and pose prior $\mathbf{P}$ to condition $\Phi$ [39, 76] to inpaint $1 - \mathbf{M}$ — the image regions that are not occupied by the human. However, there are two significant barriers to such a straightforward approach.

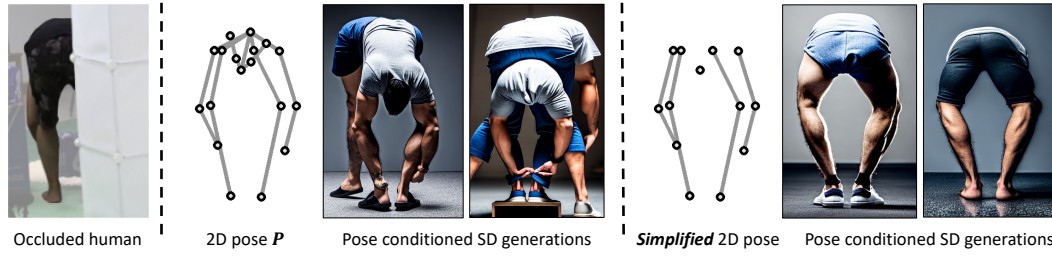

Figure 3: Stable Diffusion 1.5 generations [48] conditioned on a challenging pose $\mathbf{P}$. While conditioning on the original pose results in multiple limbs and other abnormalities, our method of simplifying pose by removing self-occluded joints results in more feasible generations.

**Conditioned human generation cannot handle challenging poses.** It is true that a conditioned diffusion prior $\Phi$ is able to generate detailed images while staying consistent with the condition. However, since diffusion models like $\Phi$ are usually overfitted on more commonly seen poses, $\Phi$ usually fails to generate reasonable images when conditioned on challenging poses (see Figure 3 middle column). We attribute this limitation to the inappropriate 2D representation of $\mathbf{P}$ — when joints occlude each other, it is impossible to tell which joints are closest to the camera when they are projected to 2D. So, we propose to simplify the 2D representation of $\mathbf{P}$. We apply a Z-buffer test on the depth map rendered from the SMPL mesh [35] and then calculate the distance $d$ between its z-axis location and the corresponding 2D z-buffer. Given a pre-defined threshold $\sigma$, we deem a joint is self-occluded if $d > \sigma$. Self-occluded joints are ignored when projecting 3D joints onto the 2D canvas for conditioning $\Phi$ (see Figure 3 right column). Our simplification improves the generation quality of $\Phi$ for challenging poses.

**Per-frame inpainting cannot guarantee cross-frame consistency.** Compared to image generation models, video generation models [11, 60, 7] are less accessible and much more expensive to run. Without an explicit modeling of object motion in the video, frame-by-frame generation with an image generative model leads to cross-frame inconsistency, which is not desirable for human reconstruction

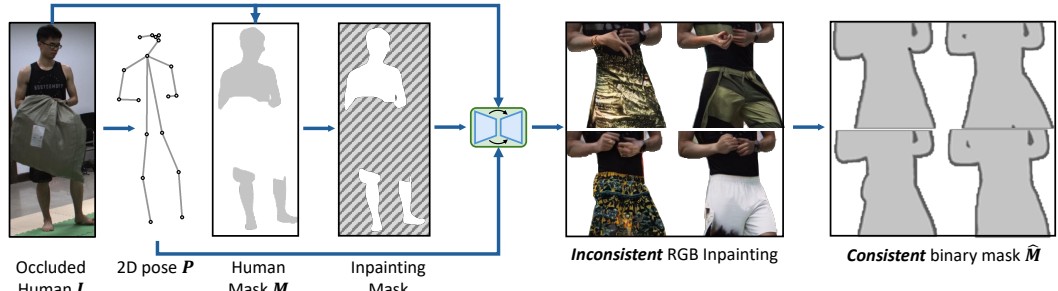

Occluded    2D pose $P$    Human    Inpainting       *Inconsistent* RGB Inpainting     *Consistent* binary mask $\hat{M}$
Human $I$            Mask $M$     Mask

Figure 4: While generative models provide inconsistent inpainting results, the binary masks that can be extracted from these generated images are much more consistent.

(see Figure 4 middle column). Instead of inpainting the occluded parts of the human directly with $\Phi$, we claim that it is more feasible to inpaint binary human masks since small variations in the human silhouette are more acceptable (see Figure 4 right column). We first inpaint the RGB image $\mathbf{I}$ and then rely on an off-the-shelf segmentation model [23] to obtain the inpainted binary human masks $\{\hat{\mathbf{M}}\}$, which is used to assist the training of the rendering model in subsequent stages.

## 4.2 Optimization Stage: Enforcing Human Completeness with SDS Regularization

After obtaining the inpainted masks $\{\hat{\mathbf{M}}\}$ that outline a reasonable human silhouette, we build a Gaussian splatting model similar to the one described in section 3.2 for human rendering. The 3D Gaussians $\Pi$ are initiated as the SMPL mesh vertices, which are able to be deformed to adapt to different poses through SMPL-based LBS (Equation 3.1). With the help of $\{\hat{\mathbf{M}}\}$, the training of $\Pi$ consists of multiple photometric loss terms $\mathcal{L}_{photo}$:

$$\lambda_{rgb}L_1(\mathbf{M}\cdot\mathbf{I}, \mathbf{M}\cdot\mathbf{I}') + \lambda_{mask}L_2(\hat{\mathbf{M}}, \mathbf{A}) + \lambda_{ssim}\texttt{SSIM}(\mathbf{M}\cdot\mathbf{I}, \mathbf{M}\cdot\mathbf{I}') + \lambda_{lpips}\texttt{LPIPS}(\mathbf{M}\cdot\mathbf{I}, \mathbf{M}\cdot\mathbf{I}'), \quad (3)$$

where $L_1$ is the L-1 loss, $L_2$ is the L-2 loss, $\texttt{SSIM}(\cdot)$ is the SSIM function [56], $\texttt{LPIPS}$ is the VGG-based perceptual loss [77], $\mathbf{I}'$ is the rendered image from $\Pi$, and $\mathbf{A}$ is the rendered human occupancy map. Each of the loss terms is scaled by a weight hyperparameter $\lambda$.

Even with the supervision of $\{\hat{\mathbf{M}}\}$, geometry inconsistency still exists. Although inconsistent human masks affect the training of $\Pi$ much less than inconsistent images, human completeness cannot be guaranteed without further steps.

**Using diffusion priors to enforce human completeness.** We build off of the insights from [53, 59, 70] and apply Score Distillation Sampling (SDS) [45] to improve the quality of human renderings and reduce artifacts. Instead of applying SDS on RGB images $\mathbf{I}'$, which causes appearance inconsistency, we apply it directly to the rendered human occupancy maps $\mathbf{A}$ so that diffusion scores are propagated to encourage complete $\mathbf{A}$:

$$\mathcal{L}_{\text{SDS}}^{(\mathbf{P})} = \mathbb{E}_{t,\epsilon}\left[w(t)\left(\epsilon_\phi(\mathbf{A}; t, \mathbf{P}) - \epsilon\right)\frac{\partial \mathbf{A}}{\partial \Pi}\right], \quad (4)$$

where $t$ is a scheduled time stamp, $w(\cdot)$ is a weighting function, $\epsilon(\cdot)$ is the UNet noise estimator in $\Phi$, and $\epsilon$ is the injected Gaussian noise.

**Using diffusion priors to regularize canonical pose.** In-the-wild videos often involve very sparse observations of the human, with only incomplete regions of the human visible in each frame. To further enforce completeness, we propose to render the human in the canonical Da-pose $\hat{\mathbf{P}}$ with the human oriented at a random angle $\in \{k\frac{\pi}{9}, k \in \mathbb{Z}\}$. Applying SDS on the canonical renderings serves as regularization and is randomly activated during training. Overall, at each training step in the Optimization Stage, the 3D Gaussians $\Pi$ are optimized towards:

$$\nabla_\Pi\left[\mathcal{L}_{photo} + \rho \cdot \lambda_{pose}\mathcal{L}_{\text{SDS}}^{(\mathbf{P})} + (1-\rho) \cdot \lambda_{can}\mathcal{L}_{\text{SDS}}^{(\hat{\mathbf{P}})}\right], \quad (5)$$

where $\rho$ is a random variable that has a 75% chance to be 1 and 0 otherwise. The Optimization stage results in a complete and coherent geometry regardless of the viewing angle.

Table 1: Quantitative comparison on the ZJU-MoCap and OcMotion datasets. LPIPS values are scaled by ×1000. We color cells that have the best and second best metric values.

| Methods | ZJU-MoCap [44] | | | OcMotion [15] | | |
|---|---|---|---|---|---|---|
| | PSNR↑ | SSIM↑ | LPIPS↓ | PSNR*↑ | SSIM*↑ | LPIPS*↓ |
| HumanNeRF [57] | $20.67^{\ddagger}$ | $0.9509^{\ddagger}$ | - | - | - | - |
| 3DGS-Avatar [46] | $17.29^{\dagger}$ | $0.9410^{\dagger}$ | $63.25^{\dagger}$ | $9.788^{\dagger}$ | $0.7203^{\dagger}$ | $188.1^{\dagger}$ |
| GauHuman [14] | 21.55 | 0.9430 | 55.88 | 15.09 | 0.8525 | 107.1 |
| OccNeRF [64] | $22.40^{\ddagger}$ | $0.9562^{\ddagger}$ | $43.01^{\ddagger}$ | 15.71 | 0.8523 | 82.90 |
| OccGaussian [67] | $23.29^{\ddagger}$ | $0.9482^{\ddagger}$ | $41.93^{\ddagger}$ | - | - | - |
| Wild2Avatar [63] | - | - | - | $14.09^{\S}$ | $0.8484^{\S}$ | $93.31^{\S}$ |
| OccGauHuman | 22.71 | 0.9492 | 54.60 | 18.85 | 0.8863 | 86.53 |
| OccFusion | 23.96 | 0.9548 | 32.34 | 18.28 | 0.8875 | 82.42 |

$^{*}$ Metrics calculated on **visible pixels** only.
$^{\dagger}$ Model trained for 5k iterations with ×3 **training time**.
$^{\ddagger}$ Results taken from OccGaussian [67], using ×5 **training frames**.
$\S$ Model trained under the default setting [63] using ×2 **training frames**.

### 4.3 Refinement Stage: Refining Human Appearance via In-context Inpainting

As shown in Figure 6 Exp. C and D, applying diffusion priors on rendered human occupancy maps is not able to recover the missing appearances of the human. This motivates the need for a subsequent stage that keeps refining $\Pi$ for better appearance.

The refinement of the appearance of 3D objects is not a new topic [53, 31, 70]. However, no existing generative models are capable of handling the consistency of appearance of a human across different frames and poses. We attribute this difficulty to the denoising process used in generative priors — random noise is injected to rendering at each SDS step which leads to uncertain results. This is infeasible for reconstruction tasks, which require frame-consistent representations that agree with all observations.

Our approach focuses on generating inpainted images of the occluded human offline to use as references. We first identify the occluded regions to be inpainted $\mathbf{R}$ by using the rendered human occupancy masks $\mathbf{A}$ from the Optimization Stage and pre-computed human visibility masks $\mathbf{M}$: $\mathbf{R} = (1 - \mathbf{M}) \cdot \mathbf{A}$. In order to encourage the generated regions to be more consistent with the partial observations, we propose in-context references inspired by in-context learning in language models [1]. Although renderings from the Optimization Stage lack sharp and high-fidelity details, they resemble complete human geometries and possess good enough features that can be used as a coarse reference to guide $\Phi$ to inpaint similar contents at occluded body regions. To achieve this, we stack $\hat{\mathbf{I}}$ and $\mathbf{I}$ together as a single image input to $\Phi$ with an additional prompt phrase — "the same person standing in two different rooms".

We use the inpainted RGB images $\{\tilde{\mathbf{I}}\}$ along with other priors to finetune $\Pi$ via photometric losses. Since diffusion models still tend to be somewhat inconsistent, we smooth training by putting more weight on perceptual loss terms and use L1 loss for the pixel-wise loss terms for its high robustness to variance:

$$\nabla_{\Pi} \left[ \lambda_{rgb} L_1(\mathbf{M} \cdot \mathbf{I}, \mathbf{M} \cdot \mathbf{I}') + \lambda_{mask} L_2(\hat{\mathbf{M}}, \mathbf{A}) + \lambda_{gen} L_1(\tilde{\mathbf{I}}, \mathbf{R} \cdot \mathbf{I}') + \lambda_{lpips} \mathtt{LPIPS}(\mathbf{I}, \mathbf{I}') \right]. \quad (6)$$

We train our entire pipeline for only **10 minutes** on a single TITAN RTX GPU. More implementation details are provided in supplementary materials.

## 5 Experiments

In this section, we conduct quantitative and qualitative evaluation of our approach against state-of-the-art methods. Then, we conduct ablation studies of our entire pipeline, demonstrating that each stage is necessary for optimal performance. **More experiments and results can be found in supplementary materials.**

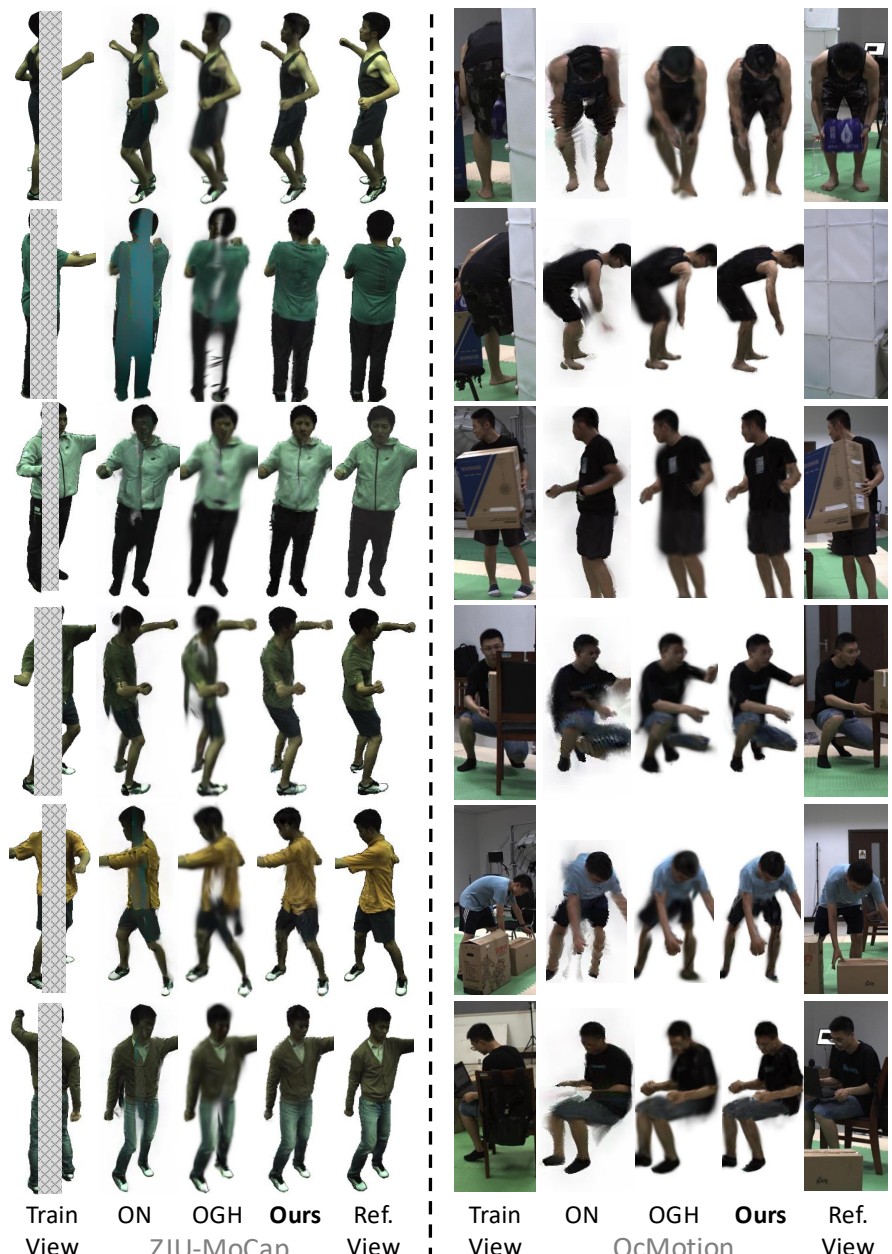

Figure 5: Qualitative comparisons on **simulated occlusions** in the ZJU-MoCap dataset [44] (left column) and **real-world occlusions** in the OcMotion dataset [16]. ON denotes OccNeRF [64] and OGH denotes OccGauHuman.

## 5.1 Datasets and Evaluation

**ZJUMoCap.** ZJU-MoCap [44] is a dataset consisting of 6 dynamic humans captured with a synchronized multi-camera system. Since the humans are in a lab environment free of occlusions, we follow OccNeRF's [64] protocol to simulate occlusion of the human, masking out the center 50% of the human pixels for the first 80 % of frames. To challenge OccFusion on videos with even sparser frames, we use only **100 frames** from the first camera with a sampling rate of 5 to train the models and use the other 22 cameras for evaluation.

**OcMotion.** OcMotion [15] comprises of 48 videos of humans interacting with real objects in indoor environments. Experiments are conducted on the same 6 sequences adopted by Wild2Avatar [63], which are selected to provide a diverse coverage of real-world occlusions. We form sparser subsequences by sampling only 50 frames from each sequence to train the models.

Table 2: Ablation results on the ZJU-MoCap [44] dataset. LPIPS values are scaled by $\times 1000$.

| Exp. | Methods | PSNR↑ | SSIM↑ | LPIPS↓ | Train time |
|---|---|---|---|---|---|
| - | GauHuman [14] | 21.55 | 0.9430 | 55.88 | 10 mins |
| A | OccGauHuman | 22.54 | 0.9457 | 54.88 | 2 mins |
| B | + *Init Stage* generated masks $\{\hat{M}\}$ | 23.52 | 0.9516 | 52.35 | 5 mins |
| C | + Posed space SDS | 23.90 | 0.9510 | 55.47 | 7 mins |
| D | + Canonical space SDS (*Optim Stage*) | 23.91 | 0.9514 | 55.35 | 7 mins |
| E | + *Refinement Stage* | **23.96** | **0.9548** | **32.34** | 10 mins |

**Evaluation.** We compare our OccFusion to OccNeRF [64], OccGaussian [67], and Wild2Avatar [63], the state-of-the-art in occluded human rendering. We also compare our results to GauHuman [14], HumanNeRF [57], and 3DGS-Avatar [46], popular human rendering methods not designed for occlusion. For fairness of comparison, all methods use the same set of segmentation masks and pose priors. We train GauHuman and OccGauHuman for 10 minutes each. We evaluate the methods both quantitatively and qualitatively. For our quantitative evaluations, we calculate the Peak Signal-to-Noise Ratio (PSNR), Structural SIMilarity (SSIM), and Learned Perceptual Image Patch Similarity (LPIPS) metrics against the ground truth images. Since no ground truth is provided for OcMotion, we calculate the metrics on visible pixels only. For qualitative evaluations, we render the human from novel views and assess the quality of the renderings.

## 5.2 Results on Simulated and Real-world Occlusions

We provide quantitative metrics averaged over all the sequences in Table 1. Overall, methods designed for occluded human rendering tend to outperform their traditional counterparts. Among those methods, OccFusion consistently performs up to par or better than the state-of-the-art on both datasets while significantly beating all the baselines on LPIPS.

Qualitative results on novel view synthesis can be found in Figure 5. OccNeRF [64] has trouble generating unseen regions and renders significant discoloration and floaters when faced with occlusion. On the other hand, OccGauHuman's renderings are blurry and occasionally incomplete. We observe that OccFusion is the only method to consistently render sharp and high-quality renderings free of occlusions.

## 5.3 Additional Studies

**Ablation studies.** We study the effect of each of our proposed components by adding them one by one and report average metrics on ZJU-MoCap in Table 2. Each stage plays a part towards optimal performance. Qualitative results on our ablations are included in Figure 6. We can see that the Initialization Stage helps enforce completeness for the initially incomplete human. The SDS regularization provided in the Optimization Stage helps remove floaters and artifacts in the posed and canonical space, further improving the shape of the human and enforcing completeness of the body. Finally, the Refinement Stage helps make the renderings more detailed in less observed regions, improving the rendering quality and greatly reducing the LPIPS.

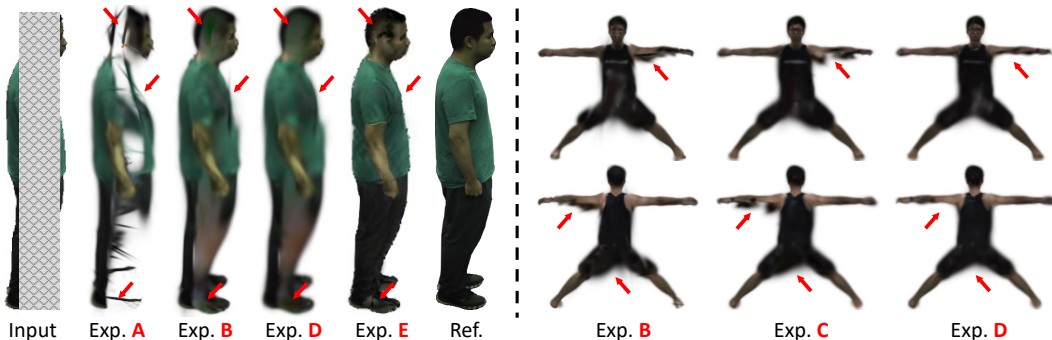

Figure 6: Qualitative ablation studies. Please see Table 2 for corresponding experiments. Major differences are highlighted by red arrows.

**Does the proposed OccGauHuman perform better than GauHuman [14] in rendering occluded humans?** In section 3.3, we present a simple upgrade for the state-of-the-art 3DGS based human rendering model GauHuman [14] to help it better handle occlusions. Our improvements are straightforward but effective. We show quantitative results in Figure 1 (Left) and Table 1. As shown in Figure 7, our improved OccGauHuman reconstructs a more complete human body than the vanilla GauHuman.

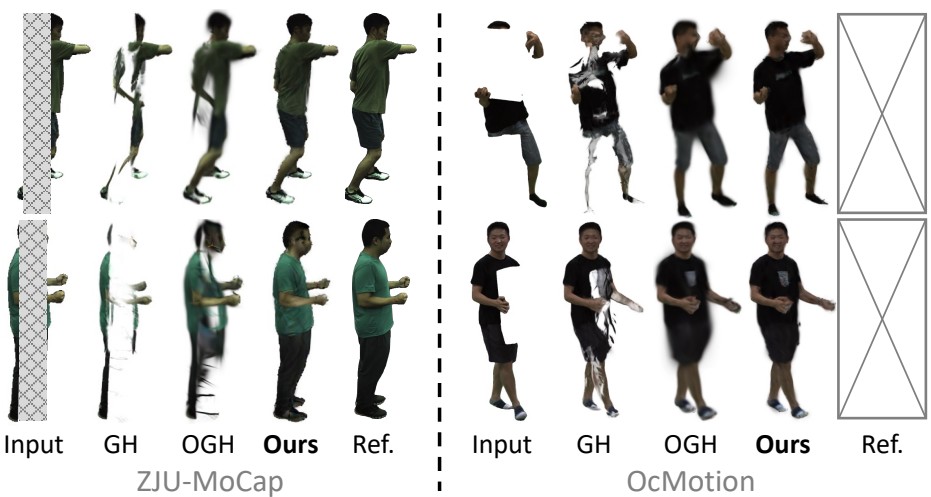

Figure 7: Qualitative comparisons on **simulated occlusions** in the ZJU-MoCap dataset [44] (left column) and **real-world occlusions** in the OcMotion dataset [16] (right column). GH denotes GauHuman [14] and OGH denotes OccGauHuman.

## 6  Discussions and Conclusion

**Limitations.** Recovering occluded dynamic humans is challenging. As mentioned in section 4.3, reconstructing a 3D human requires adhering to multiple consistencies. However, even with the state-of-the-art generative models, it is still impossible to perfectly maintain those consistencies for 4D content (3D + motion) generation. Although our proposed methods are specifically designed to eliminate potential variances when using generative priors, we can still observe some generations are less coherent (e.g. Figure 4 and Figure 8), which may hurt the training of the rendering model on all stages. Moreover, we found that conditioning generative models with 2D poses is weak — the pose of the generated human does not always align with the condition pose, which may introduce even more uncertainty for training. In future work, we hope to train our own consistency-aware diffusion model specifically finetuned on human data.

**Societal Impacts.** Being able to reconstruct a human from an occluded monocular video can have a great societal impact. For example, having a high-fidelity 3D reconstruction of a human can help telemedicine practitioners become more immersed in the 3D space. While our research could lead to privacy concerns if humans are reconstructed without their consent, we believe that the benefits can be harnessed responsibly with appropriate safeguards.

**Conclusion.** In this work, we propose OccFusion, one of the first works that utilize 3D Gaussian splatting for occluded human rendering. Our approach consists of three stages: the Initialization, Optimization, and Refinement stages. By combining the efficiency and representative ability of 3D Gaussian splatting with the generation capabilities of diffusion priors, our method achieves state-of-the-art in occluded human rendering quality as measured by the PSNR, SSIM, and LPIPS metrics while only taking around 10 minutes to train. We hope our work inspires further exploration into the capabilities of diffusion priors to aid in human reconstruction.

# 7 Acknowledgment

This work was partially funded by the NIH Grant R01AG089169 and P41EB027060, Panasonic Holdings Corporation, the Gordon and Betty Moore Foundation, the Jaswa Innovator Award, Stanford HAI, Stanford HAI graduate fellowship, and Stanford Wu Tsai Human Performance Alliance.

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

# A  Table of Symbols

For notation simplicity, we adopted alphabetic symbols in this paper to represent essential components in our framework. For better symbol-name correspondences, here we justify the implications of all symbols used in the paper in Table 3.

Table 3: Table of symbols.

| Symbols | Explanations |
|---|---|
| *Preliminaries* | |
| $\mathbf{x_c}$ | 3D points in the canonical human space |
| $\mathbf{x_P}$ | 3D points in the posed human space |
| $w$ | skinning weights used in LBS |
| $G$ | transformation matrix used in LBS |
| $b$ | translation vector used in LBS |
| $\mathbf{J}$ | 3D locations of human joints |
| $\theta$ | pose parameters used in SMPL [35] |
| $\beta$ | shape parameters used in SMPL [35] |
| $\mathbf{p}$ | center of a 3D Gaussian |
| $o$ | opacity of a 3D Gaussian |
| $\mu$ | mean value of a 3D Gaussain |
| $\Sigma$ | covariance matrix of a 3D Gaussain |
| *OccFusion* | |
| $\Pi$ | optimizable human 3D Gaussians |
| $\Phi$ | a pretrained generative model [48], used as prior |
| $\mathbf{M}$ | precomputed binary human mask, used as prior |
| $\mathbf{P}$ | precomputed human pose, used as prior |
| $\hat{\mathbf{P}}$ | the canonical articulation of $\mathbf{P}$ |
| $\mathbf{I}$ | input image with occluded human |
| $\hat{\mathbf{M}}$ | Init Stage generated complete human mask |
| $\Delta$ | SDS gradients, used as a guidance in the Optim Stage |
| $\hat{\mathbf{I}}$ | Optim Stage rendered human RGB image |
| $\mathbf{A}$ | $\Pi$ rendered human occupancy map in all stages |
| $\mathbf{C}$ | Refine Stage rendered human RGB image |
| $\mathbf{R}$ | inpainting mask computed by $(1 - \mathbf{M}) \cdot \mathbf{A}$ |
| $\rho$ | a random variable $\in [0, 1]$ controls Optim Stage SDS |

# B  Implementation Details

OccFusion requires several priors. We run SAM [25] to get all the human masks $\{\mathbf{M}\}$. While we follow previous work [64, 63] and use the ground truth poses provided by ZJU-MoCap and OcMotion, pose priors $\mathbf{P}$ can be obtained via occlusion-robust SMPL prediction/optimization methods such as HMR 2.0 [8] and SLAHMR [69] for in-the-wild videos. Improving the quality of priors is not the focus of this work. We use the pre-trained Stable Diffusion 1.5 model [48] with ControlNet [76] plugins for SDS in all the stages.

In the *Initialization Stage*, instead of inpainting incomplete human masks directly, we run the pretrained diffusion model to inpaint RGB images with 10 inference steps and 1.0 ControlNet conditioning scale. We use the positive prompt — *"clean background, high contrast to the background, a person only, plain clothes, simple clothes, natural body, natural limbs, no texts, no overlay"* and the negative prompt — *""multiple objects, occlusions, complex pattern, fancy clothes, longbody, lowres, bad anatomy, bad hands, bad feet, missing fingers, cropped, worst quality, low quality, blurry"*. After inpainting the RGB images, we then run SAM-HQ [23] with $\mathbf{P}$ as the prompts to get $\{\hat{\mathbf{M}}\}$.

In the *Optimization Stage*, we train the 3D human Gaussian $\Pi$ from scratch by following the objective Equation 5. We set $\lambda_{rgb} = 1e^4$, $\lambda_{mask} = 2e^4$, $\lambda_{ssim} = 1e^3$, and $\lambda_{lpips} = 1e^3$. At each training step, we random switch the SDS regularization on either posed human space or the canonical Da-pose space with a probability of 75% and 25%. When applying SDS regularization on the canonical human

space, we randomly rotate the human horizontally with a uniformly sampled degree in $\{k\frac{\pi}{9}, k \in \mathbb{Z}\}$. We set the SDS loss weights as $\lambda_{pose} = 2e^5$ and $\lambda_{can} = 2e^5$. In this stage, we train $\Pi$ for 1200 steps.

In the *Refinement Stage*, we first generate the RGB human inpaintings via the proposed in-context inpainting method. We run the pretrained diffusion model with conditions on $\mathbf{M}$, 10 inference steps, and 0.3 ControlNet conditioning scale. We did not use positive prompts for the inpainting but used the same negative prompts as in the *Optimization Stage*. During training, we set the loss weights as $\lambda_{rgb} = 1$ and $\lambda_{mask} = 0.1$, $\lambda_{gen} = 0.1$, and $\lambda_{lpips} = 0.2$. In this stage, we finetune $\Pi$ for another 1800 steps with Gaussian densification and pruning enabled for the first 1000 steps.

## C  Additional Studies

**Effectiveness of in-context inpainting.** We provide comparisons of the human in the Refinement Stage with and without in-context inpainting and provide qualitative comparisons in Figure 8. While renderings from the Optimization stage are less detailed in occluded areas, our proposed in-context inpainting is able to generate the missing content and greatly increase the rendering quality in these areas.

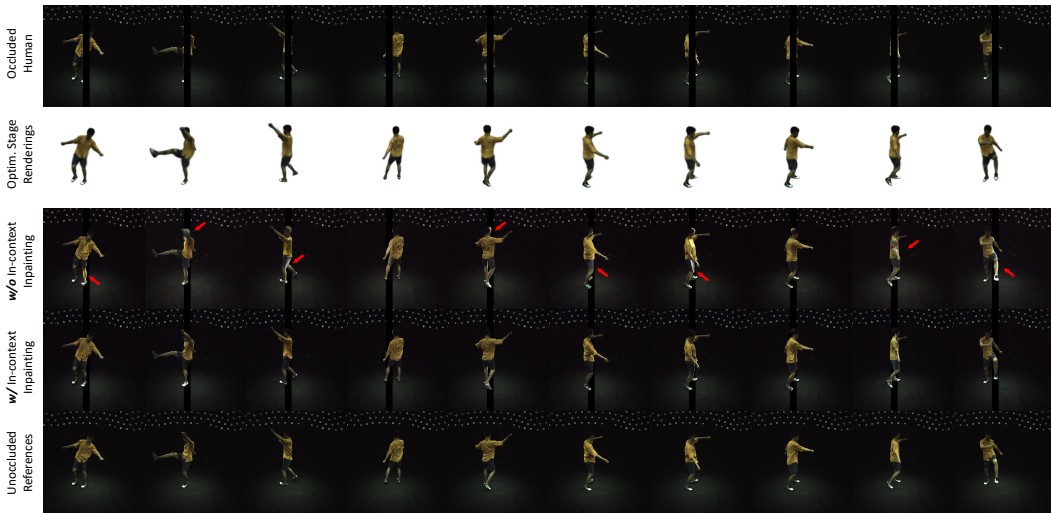

Figure 8: Comparison of the inpainted human in the Refinement Stage with and without using the proposed in-context inpainting technique. Major differences are highlighted with red arrows.

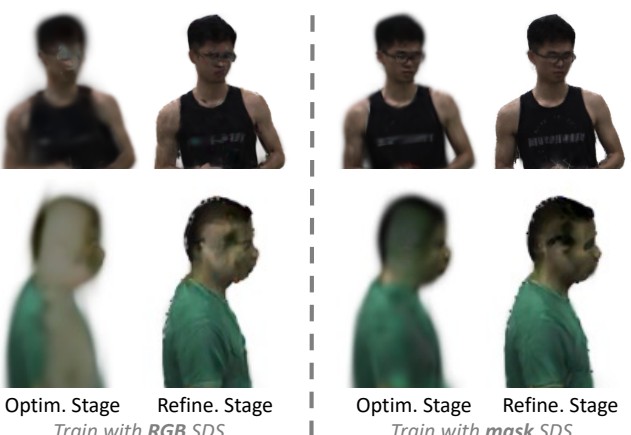

Figure 9: Applying SDS on RGB images vs. on human occupancy maps. As mentioned in Sec. 4.1 and Fig. 4 of the main paper, generated RGB appearances are much more inconsistent than generated silhouettes. As a result, applying SDS on RGB leads to defective rendering results.

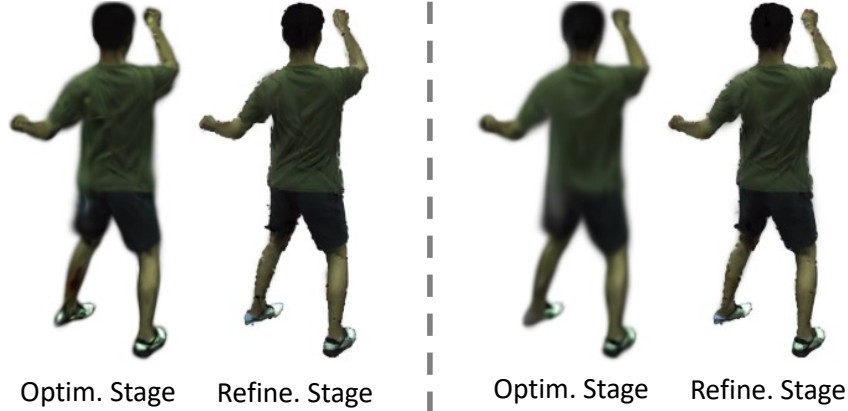

Optim. Stage    Refine. Stage        Optim. Stage    Refine. Stage

*Train with **complete** masks*      *Train with **in-painted** masks*

Figure 10: Training with complete unoccluded masks vs. with inpainted masks in the Optim. stage. Although inpainted masks are slightly more inconsistent compared to the complete masks, our training pipeline converges to the same level of rendering quality.

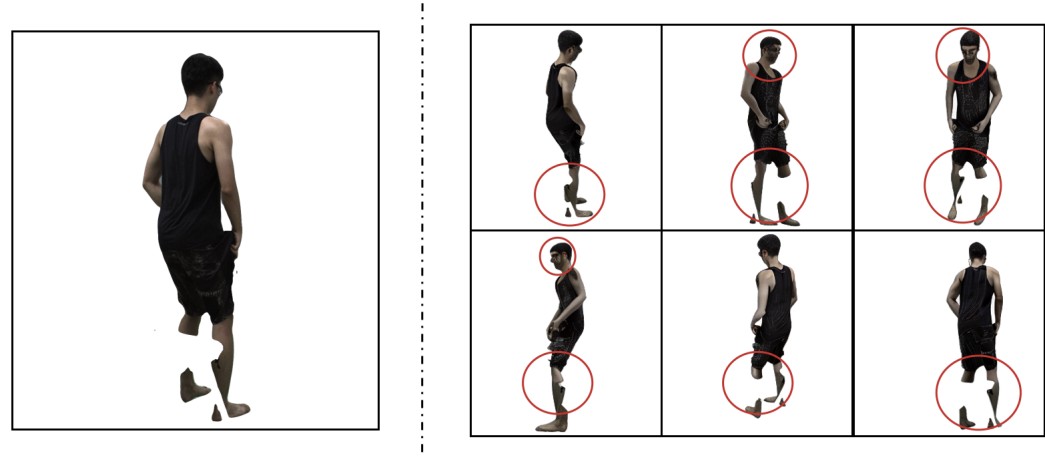

Input image condition                    Novel view synthesis

Figure 11: Novel view synthesis results from InstantMesh [65] conditioned on the least occluded frame. Discrepancies are circled in red.

**Applying SDS on RGB vs on Human Occupancy Maps.** We include additional experiments comparing the rendering results of applying SDS on RGB vs. on human occupancy maps (as proposed). It is clear that applying SDS on RGB leads to defective renderings as well as inferior quantitative results. This experiment validates our claim made in Sec. 4.1 and Fig. 4 in the main paper.

**Robustness of training to inpainted masks.** For in-the-wild occluded videos, there are no ground truth masks for the occluded body regions due to unknown human/garment deformations. Relying on the state-of-the-art pre-trained priors brought by the Segment Anything model (SAM) [25] and Stable Diffusion [48], the segmented/inpainted masks are expected to be reasonable and coherent across frames. To test the robustness of our method to variances in the in-painted masks, we add comparison experiments on ZJU-MoCap that supervise using the complete SAM masks obtained from the unoccluded humans with minimum variances. Please see the qualitative results in Figure 10. We find that using the inpainted $\hat{M}$ leads to a good enough rendering quality comparable to using masks derived from the unoccluded images, validating the robustness of our model.

**Can existing generative models recover an occluded human?** While there are works for using generative diffusion models to render 3D humans conditioned on single [58] and multiple [49] images, none are able to condition on a monocular video of the person.

Since [58] has not released code, we include results from InstantMesh [65]. We use the provided segmentation mask to mask the least occluded frame onto a white background and use it as conditioning. Novel view synthesis results are included in Figure 11. InstantMesh is unable to recover a complete human geometry and fails to generate a reasonable appearance from the single image.

## D   Video Studies

For a more comprehensive presentation of the results, we include video renderings on all the training frames for both datasets. For the ZJU-MoCap videos (named with the prefix `zju`), from left to right, we show the occluded human, OccGauHuman rendering, Optimization Stage rendering, Refinement Stage rendering, and the reference. For the OcMotion videos (named with the prefix `ocmotion`), without references for real-world occlusions, from left to right, we show the occluded human, OccGauHuman rendering, Optimization Stage rendering, and Refinement Stage rendering.

