# OpenReview forum: "OccFusion: Rendering Occluded Humans with Generative Diffusion Priors"
_NeurIPS.cc/2024/Conference — NeurIPS 2024 poster_

### Official Review · Reviewer_BvJA · 2024-06-13

**Soundness:** 3
**Presentation:** 2
**Contribution:** 2
**Rating:** 5
**Confidence:** 5

**Summary:**

This paper presents OccFusion, a 3D human avatar creation system that combines 3D Gaussian Splatting (3DGS) and 2D diffusion models to effectively render occluded regions. To this end, three stages are designed. First, in the initialization stage, complete human masks (without occluder) are obtained using a pre-trained diffusion model. Then, in the optimization stage, 3DGS is optimized with score distillation sampling (SDS) to recover occluded regions. Finally, in the refinement stage, in-context inpainting is designed to improve the rendering quality.

**Strengths:**

What the authors try to address (recovering 3D human avatar from human images with occlusions) is an important and unexplored problem. The authors’ design, which incorporates pre-trained diffusion models, is a reasonable choice.

**Weaknesses:**

1. Overall, the rendering quality is bad. This is applicable even for visible areas. Fig. 5 and 6 shows that the results are quite blurry especially on the OcMotion dataset.

2. I don’t think the rightmost binary masks in Fig. 4 are consistent. All of them obviously have different pants silhouettes. Given these inconsistent masks, which masks did the authors use to optimize 3DGS?

3. Overall, paper writing should be polished.

--- post rebuttal ---

The authors addressed my concerns well. I'm raising my rating.

**Questions:**

1. What makes the result look so blurry even for visible areas?
2. In Fig. 5, the second column and second row results do not have visible humans in the reference.
3. How much of the mask (M) and complete human mask (\hat{M}) are accurate? Also, how much of the optimization stage is robust to wrong M and \hat{M}?
4. Fig. 3 shows a pose-to-image generation example, while the actual use case is (image+pose+mask)-to-image as in Fig. 4. In the case of Fig. 4, the pose simplification of FIg. 3 is still necessary?
5. Fig. 5 shows that the leftmost column is an input. What do the authors mean by input? As far as I understand, 3DGS is optimized to replicate images of a single person in the training set, and there is no input to the 3DGS. Or the 3DGS takes images of any person and can render images of that person with novel poses and viewpoints?

**Limitations:**

The major limitation is the rendering quality. Most of the rendered images are quite blurry.

---

> ### Author Rebuttal · Authors · 2024-08-07
>
> We thank the reviewer for your time and the helpful comments! We address your concerns below.
>
> > Rendering quality seems blurry
>
> With only 10 mins of training time, OccFusion surpasses state-of-the-art occluded human rendering methods by a significant margin qualitatively and quantitatively, as illustrated in Table 1 and Fig. 1 and 5 of the main paper. With that said, with a longer training time, OccFusion is able to render in higher quality with much less blur. We added supplementary experiments to validate this statement – please see qualitative and quantitative results in Fig. 1 in the rebuttal PDF. We believe that our proposed 10-minute version of OccFusion is able to achieve the best balance of rendering quality and efficiency.
>
> > Is $M$ accurate?
>
> As mentioned in the implementation details section of our Appendix, we derive $M$ from SAM (ICCV 2023), which is among the state-of-the-art in image segmentation, making it a very reliable way to calculate $M$. We would like to note that numerous other human rendering works [1, 2, 3] also use SAM as their go-to method for obtaining pseudo ground truth human masks during preprocessing.
>
> [1] Kocabas, Muhammed, et al. "Hugs: Human gaussian splats." Proceedings of the IEEE/CVF conference on computer vision and pattern recognition. 2024.
>
> [2] Pang, Haokai, et al. "Ash: Animatable gaussian splats for efficient and photoreal human rendering." Proceedings of the IEEE/CVF Conference on Computer Vision and Pattern Recognition. 2024.
>
> [3] Hu, Liangxiao, et al. "Gaussianavatar: Towards realistic human avatar modeling from a single video via animatable 3d gaussians." Proceedings of the IEEE/CVF Conference on Computer Vision and Pattern Recognition. 2024.
>
>
> > Binary masks do not seem consistent — Is $\hat{M}$ accurate?
>
> When dealing with in-the-wild data, the ground truth silhouette of occluded regions is always unknown. Since the silhouette of the human across different frames is supposed to be inconsistent due to human (and sometimes garment) movement, we will never know with certainty what the changes in the ground truth silhouettes would be across a video sequence. However, we find that by using a pre-trained Stable Diffusion prior that has been trained on a great amount of similar data with enforced consistency, we can ensure that the inpainted $\hat{M}$ falls into a reasonable estimation given the context of the pose and visible regions. As Fig. 4 of the main paper demonstrates, the shape of inpainted binary masks is much more reliable and consistent than the appearance of the corresponding RGB inpainted images. In our pipeline, we generate only one mask for every frame and find that $\hat{M}$ does well in supervising the training of later stages.
>
> > Robustness of Optimization Stage to wrong $M$ and $\hat{M}$
>
> As we covered in the past two responses, the fact that we utilize pre-trained SAM to get $M$ and pose-conditioned pre-trained Stable Diffusion to inpaint $\hat{M}$ makes getting a “wrong” mask extremely unlikely. However, to test the robustness of our method to inpainted masks, we add experiments on ZJU-MoCap supervised on the complete SAM masks obtained from the **unoccluded humans** in the optimization stage. Please see the qualitative and quantitative results in Fig. 2 in the rebuttal PDF. We find that using the inpainted $\hat{M}$ can lead to good rendering quality that is comparable to using masks derived from the unoccluded images, validating that our model is robust to variances that exist in the masks.
>
> > Is pose simplification necessary for image + pose + mask -> image?
>
> The purpose of Figure 3 is to show that Stable Diffusion fails when conditioned on challenging 2D poses. Since we use the same model for the image inpainting process (with the masks being used to calculate the necessary regions for inpainting), the pose simplification step we proposed is still necessary for the Stable Diffusion model to accurately condition on the pose of the human. So, we apply the pose simplification step whenever self-occluded joints are present to discourage problematic generations.
>
> > Fig. 5 shows that the leftmost column is an input. 3DGS is optimized to replicate images of a single person in the training set, and there is no input to the 3DGS.
>
> This is correct. The leftmost columns are not really inputs but reference images at the corresponding time stamp from the training point of view. We apologize for this error and will correct the figure to clarify this point.
>
> > Fig 5 Second Column Second Row does not have a visible human in reference
>
> In Figure 5, we validate renderings from novel views. The reference images on the right show the views from the new camera perspective that is provided by the dataset. For an in-the-wild dataset like OcMotion, sometimes the camera cannot capture the complete human due to occlusions. In the case of the example on row 2 column 2, the human is completely occluded from the reference camera view. We will make this point more clear in the final version.

---

> > ### Comment · Reviewer_BvJA · 2024-08-11
> >
> > Thanks for your rebuttal. How the results become if the authors just use ZJU-Mocap as it is without simulating the occlusions? I want to know the rendering capability of the proposed method if there is no occlusion as the updated results (from longer training) still do not have sharp textures.

---

> > > ### Author Response · Authors · 2024-08-12
> > > **Reply to Reviewer's Comment**
> > >
> > > Thank you for your thorough review and for considering our rebuttal.
> > >
> > > We would like to clarify that OccFusion employs the same rendering technique as introduced in GauHuman (CVPR 2024). In scenarios where there are no occlusions in the video sequence, OccFusion functions similarly to GauHuman, with the only addition being the opacity SDS in the Optimization stage.
> > >
> > > As per your request, we conducted additional experiments to assess the capacity of OccFusion for rendering high-fidelity details. **Although figures cannot be included in this response**, we present the following quantitative comparisons after training both OccFusion and GauHuman for 10 minutes on a black background:
> > >
> > > | Method       | PSNR  | SSIM   | LPIPS  |
> > > |--------------|-------|--------|--------|
> > > | GauHuman     | **31.44** | **0.9650** | 30.48  |
> > > | **OccFusion**| 31.29 | 0.9627 | **30.40** |
> > >
> > > Both quantitative and qualitative results indicate that OccFusion performs on par with GauHuman, with even better LPIPS, which often indicates of perceptual quality of renderings and **sharpness of textures**.
> > >
> > > We appreciate the reviewer’s concern and will include both the quantitative and qualitative results in the camera-ready version.

---

> > > > ### Author Response · Authors · 2024-08-13
> > > >
> > > > We would like to thank the reviewer once again for your helpful feedback and suggestions. We hope that our additional experiments and responses have addressed your questions. We would be happy to continue the discussion if the reviewer has additional concerns.

---

### Official Review · Reviewer_WHSE · 2024-06-22

**Soundness:** 3
**Presentation:** 3
**Contribution:** 3
**Rating:** 7
**Confidence:** 3

**Summary:**

This paper introduces OccFusion, a method for rendering occluded humans.

Similar to other 3DGS-based human rendering methods, OccFusion optimizes a set of 3D Gaussians to improve training and rendering speed.

OccFusion proposes adopting generative diffusion priors to ensure complete and high-quality renderings to aid in reconstruction.

OccFusion was evaluated on ZJU-MoCap and challenging OcMotion sequences, and it achieved state-of-the-art performance in rendering occluded humans.

**Strengths:**

The 3-stage strategy is efficient in handling occluded humans.

During the initialization stage, inpaint occluded human visibility masks into complete human occupancy masks.

During the optimization stage, initialize a set of 3D Gaussians and optimize them by SDS in both the posed and canonical space.

During the refinement stage, inpaint unobserved regions of the human with context from partial observations and renderings from the previous stage.

This method outperforms the state-of-the-art in rendering humans from occluded videos.

**Weaknesses:**

Lack of discussion about NeRF in the Wild, Ha-NeRF, Gaussian in the Wild and NeRF On-the-go.

**Questions:**

It is better to include more video results and add diversities of datasets.

**Limitations:**

The authors adequately addressed the limitations.

---

> ### Author Rebuttal · Authors · 2024-08-07
>
> We thank the reviewer for the positive assessment of our work and the helpful comments! We address your concerns below.
>
> > Lack of discussion about NeRF in the Wild, Ha-NeRF, Gaussian in the Wild and NeRF On-the-go
>
> Thanks for the suggestion. We will add the following sentences to section 2.2 of our Related work section in our final version:
>
> ```
> NeRF-W [1] and other works [2,3,4] are able to account for photometric variation and transient occluders in complex in-the-wild scenes, allowing them to render consistent representations from unconstrained image collections.
> However, these works are not designed to handle dynamic objects like humans.
> ```
>
>
>
> [1] Martin-Brualla, Ricardo, et al. "Nerf in the wild: Neural radiance fields for unconstrained photo collections." Proceedings of the IEEE/CVF conference on computer vision and pattern recognition. 2021.
>
> [2] Chen, Xingyu, et al. "Hallucinated neural radiance fields in the wild." Proceedings of the IEEE/CVF Conference on Computer Vision and Pattern Recognition. 2022.
>
> [3] Ren, Weining, et al. "NeRF On-the-go: Exploiting Uncertainty for Distractor-free NeRFs in the Wild." Proceedings of the IEEE/CVF Conference on Computer Vision and Pattern Recognition. 2024.
>
> [4] Zhang, Dongbin, et al. "Gaussian in the Wild: 3D Gaussian Splatting for Unconstrained Image Collections." arXiv preprint arXiv:2403.15704 (2024).
>
>
> > More visual results
>
> We provide more experiment results in the PDF of the general rebuttal, and we will include more extensive video results in the final version.
>
> > Diversity of dataset
>
> This is a great observation. A big restriction of current human rendering datasets (ZJU-MoCap, OcMotion) is that they lack diversity. ZJU-MoCap consists of humans rotating in place in a brightly lit motion capture environment. While OcMotion is more representative of a real world scene, its diversity is still lacking, with all the sequences being collected in the same indoor room. In addition, since both datasets are collected from Chinese universities, the subjects are all East Asian men. We believe that a promising future step for this field is to collect more diverse data to test the generalizability of human rendering methods like ours. We will add a discussion of dataset diversity to our Limitations section in the final version.

---

> > ### Comment · Reviewer_WHSE · 2024-08-11
> >
> > Thanks for the rebuttal. It addressed my concerns well.

---

### Official Review · Reviewer_Vqeb · 2024-07-09

**Soundness:** 3
**Presentation:** 3
**Contribution:** 3
**Rating:** 7
**Confidence:** 3

**Summary:**

This paper proposes a method for reconstructing gaussian-based human avatars from occluded captures. The gaussian avatar model is based on GauHuman, which is optimized in multiple stages, including using a diffusion based prior in the canonical space to recover the complete human. In the first stage, a consistent human silhouette is generated using "z-buffered" pose key points. Next, this silhouette is filled in with textures using SDS which are then further refined by a diffusion model (same as one used for SDS) using the phrase - "the same person standing in two different rooms". Finally, these inpainted images are used to refine the color of the gaussians to generate the final avatar. Both quantitative and qualitative results demonstrate the effectiveness of the proposed method over prior work. While each component of the model is not novel in itself, the overall method is, and it works well.

**Strengths:**

1) The use of the diffusion model to generate the silhouette, inpaint it in the canonical space and use it as pseudo ground-truth is novel.
2) The architecture is well motivated, with the purpose of each component very intuitive and well ablated.
2) The evaluations seem thorough for the most part, apart from the ablations.
3) The paper is well written.

**Weaknesses:**

1) In Figure 6, it would be nice to show results on both subjects for all experimental configurations (Exp A to E). In its current form it is incomplete.

**Questions:**

1) I'm curious as to whether the authors tried inpainting in the deformed space as well? If yes, does it work better or worse. It will be great if the authors could include those results as well

**Limitations:**

Yes

---

> ### Author Rebuttal · Authors · 2024-08-07
>
> We thank the reviewer for your time and the helpful comments! We address your concerns below.
>
> > Incompleteness of Figure 6
>
> Thanks for the suggestion. The reason why we do not include results for all experiments for both subjects in Figure 6 is to showcase the benefits of our proposed components on the canonical and deformed spaces. We aim to show improvements in rendered appearance in the deformed space with the subject on the left, and aim to show how human completeness is improved and artifacts are reduced in the canonical space with the subject on the right. We include more visual experiment results in the rebuttal PDF.
>
> > Inpainting in deformed space
>
> In the Optimization Stage, we apply Score Distillation Sampling (SDS) to both the deformed and canonical spaces (see Sec. 4.2 and Eq. 5 of the main paper). The space is controlled by a random variable with a 75% probability of applying SDS on the deformed space and 25% on the canonical space. We add additional experiments to show that our proposed random mix of deformed/canonical space SDS yields the best results compared to applying SDS solely on the deformed space or the canonical space (please see Fig. 4 in the rebuttal PDF).
>
> In the Refinement Stage, we operate solely in the deformed space – we render posed human renderings from the Optimization Stage and juxtapose them with occluded partial observations from the input video in order to perform our in-context inpainting.

---

### Official Review · Reviewer_9Cmr · 2024-07-11

**Soundness:** 4
**Presentation:** 3
**Contribution:** 4
**Rating:** 6
**Confidence:** 4

**Summary:**

OccFusion proposes an approach to model human bodies that fail under occlusion in monocular videos. The authors utilize 3D Gaussian splatting for efficient rendering and leverage pretrained off-the-shelf image diffusion models as 2D priors. Their approach involves a three-stage training process, sequentially refining estimates to achieve precise 3D human representations.  They demonstrate superior performance compared to previous methods, enhancing one of the baselines to better handle occlusion scenarios.

**Strengths:**

1. The paper is well-written and provides valuable engineering insights, particularly through the extensive use of various off-the-shelf models. The authors demonstrate proficient application of these methods to their tasks.

2. As for another strength, they enhance the GauHuman baseline to handle occlusion correctly, a significant contribution worth highlighting.

3. The ablation study is informative and visual, effectively supporting the narrative.

4. The qualitative study shows the method's superiority, with well-justified reasons in the text.

**Weaknesses:**

1. Some important details are either omitted or mentioned too late in the text. Below are some examples:

1.1. While the extensive use of off-the-shelf models is well justified, it is only superficially discussed in the corresponding technical sections. For instance, the introduction does not mention the use of image generative models at the initialization stage to improve the masks. This is a core aspect of the work, and mentioning models like DreamFusion, Stable Diffusion, and ControlNet should be done when discussing the three stages, preferably in the introduction.

1.2. Figure 2 (method overview) is cumbersome and not very instructive. It does not clarify the three-stage training, and its understanding relies heavily on the detailed ablation study. I recommend reorganizing Figure 2 to include all off-the-shelf models used in the training pipeline. Even a plain list of inputs and outputs with the side models used at each stage would be more descriptive than the current Figure 2 and the introductory motivation.


2. Obscure training details

2.1 Training time: authors insist throughout the text that the training time of their method is 10 minutes. Assuming the monocular video is the only input, the result must depend on the length of the video, yet it is never discussed.

2.2 It is unclear why 10 minutes is enough for good performance. Would estimates improve if trained longer?


3. Results

3.1 In Fig.6 Exp.A authors demonstrate that OccGauHuman estimate has some bodyparts ripped apart. This should not happen if initial splats are distributed all over SMPL mesh and "densification and pruning of splats is disabled. (line 137)". How can authors explain this behavior?

3.2 In Table 2 GauHuman and OccGauHuman are trained for 10 and 2 minutes, correspondingly. This is unclear how estimates improve when OccGauHuman is trained for 10 minutes, as GauHuman.
3.3 Continuing the issue 3.2, if all experiments with OccGauHuman model are also taken with 2-min limit, this makes the quantitative comparison unfair. maybe it is simply undertrained.


4. (minor) OccGauHuman. In Sec.3 authors should say explicitly that this method they propose is not the part of the contribution but the way to build a better baseline and a starting point for their own method.

5. (minor) Throughout the paper authors say "gaussian" referring to the splat or blob. It is better to avoid such namedropping, since Gaussian is a very broad term that can imply many different things and is itself an adjective and does not fit in the sentence properly.

**Questions:**

I would like authors to answer to the issues I raise in Weaknesses. If I find them constructive, I would raise the rating.

**Limitations:**

-

---

> ### Author Rebuttal · Authors · 2024-08-07
>
> We thank the reviewer for your time and the helpful comments! We address your concerns below.
>
> > Discussion of off-the-shelf models
>
> Thanks for the suggestion. We will include the following edited text in the Introduction of the final version:
>
> ```
> In this work, we introduce OccFusion, an efficient yet high quality method for rendering occluded humans. To gain improved training and rendering speed, OccFusion represents the human as a set of 3D Gaussians. Like almost all other human rendering methods, OccFusion assumes accurate priors such as human segmentation masks and poses are provided for each frame, which can be obtained with state-of-the-art off-the-shelf estimators such as SAM and HMR 2.0. However, to ensure complete and high-quality renderings under occlusion, OccFusion proposes to utilize generative diffusion priors, more specifically pose-conditioned Stable Diffusion 1.5 with ControlNet plugins, to aid in the reconstruction process.
> ```
>
> > Instructiveness of Figure 2
>
> Thanks for your suggestions. We will carefully revise Figure 2 to incorporate your comments.
>
> > Effect of length of video on training time
>
> The training time of the video does indeed depend on the length of the video. As mentioned in section 5.1 of the paper, we train our model on 100 frames for ZJU-MoCap and 50 frames for OcMotion. We also show in Figure 1 that our model outperforms some baselines that require much more (>500) frames and much longer training time. With longer training time and/or more samples from the videos, OccFusion is able to provide even better results (see Fig. 1 in the rebuttal PDF).
>
> > Does training longer improve rendering results?
>
> Yes. With only 10 mins of training time, OccFusion already surpasses all counterparts by a significant margin qualitatively and quantitatively. It is also true that with longer training time, OccFusion is able to render higher quality. We add experiments to validate this statement, please see the qualitative and quantitative results in Fig. 1 in the rebuttal PDF. We believe that our proposed 10 minute version of OccFusion is able to achieve the best balance of rendering quality and efficiency.
>
> > OccGauHuman has splats that are “ripped apart”
>
> Although the number of 3D Gaussians does not change during training, as densification and pruning of splats are disabled, the mean, scale, and opacity of each 3D Gaussian is optimized and changed during training, resulting in the “ripped apart” appearance when rendered in 2D. This is because 3D Gaussians are crowded in the visible regions. This ripped apart appearance only occurs in occluded areas, demonstrating that OccGauHuman alone is unable to reconstruct unobserved areas due to its lack of generative capabilities.
>
> > Training time of OccGauHuman
>
> The OccGauHuman model we use to compare to GauHuman and OccFusion in Table 1 is trained for 10 minutes. We will be sure to clarify this in the final version.
>
> It is also worth noting that GauHuman and OccGauHuman do not benefit significantly from training for longer. We show in Fig. 1 of the main paper that increasing the training time from 2 to 10 minutes causes only a small boost in PSNR on ZJU-MoCap for OccGauHuman and actually causes a degradation in performance for GauHuman. Additional results in Fig. 1 of the rebuttal PDF also validate this.
>
> > OccGauHuman is not a contribution
>
> Thanks for the suggestion. We will rephrase Sec 3.3 of our paper to clarify that we do not consider OccGauHuman a contribution but rather a better baseline.
>
> > Gaussians are a broad term
>
> Thanks for the suggestion. We will follow current Gaussian splatting literature [1,2] and replace all mentions of “Gaussians” in our paper with the term “3D Gaussians”.
>
>
>
>
> [1] Kocabas, Muhammed, et al. "Hugs: Human gaussian splats." Proceedings of the IEEE/CVF conference on computer vision and pattern recognition. 2024.
>
> [2] Hu, Shoukang, Tao Hu, and Ziwei Liu. "Gauhuman: Articulated gaussian splatting from monocular human videos." Proceedings of the IEEE/CVF Conference on Computer Vision and Pattern Recognition. 2024.

---

> > ### Comment · Reviewer_9Cmr · 2024-08-09
> > **Reply to Rebuttal**
> >
> > I thank authors for their instructive and well detailed answer. All of the issues I raised are well addressed.

---

> > > ### Author Response · Authors · 2024-08-09
> > >
> > > We thank the reviewer very much for your positive feedback on our rebuttal! We are glad that that we have effectively addressed the concerns you raised in your initial review,
> > >
> > > Given that you mentioned in your original review that you would consider raising your score if our rebuttal was constructive, we would like to kindly ask the reviewer to reconsider their score. We would also be happy to engage further during the discussion period.

---

### Official Review · Reviewer_gVDG · 2024-07-12

**Soundness:** 3
**Presentation:** 3
**Contribution:** 2
**Rating:** 5
**Confidence:** 5

**Summary:**

This work presents a Gaussian-based approach to reconstruct 3D human poses from occluded monocular video. Building upon the success of previous methods like SDS, the paper suggests introducing a pre-trained diffusion prior to complete the occluded areas and further refine the missing appearance. The framework is evaluated on public datasets, including ZJU-MoCap and OcMotion, and shows improved efficiency and appearance quality.

**Strengths:**

- This paper tackles a relatively new and practical problem of 3D human reconstruction from occluded monocular video.
- The introduction section provides a clear motivation and background for the proposed method.
- The authors conduct comprehensive experiments to validate the efficiency of each component of the framework.

**Weaknesses:**

- The ablation study could be more extensive. The authors mention that "applying SDS on RGB images causes appearance inconsistency" in line 192. It would be helpful if the authors could provide some visualizations to support this claim.
- If the optimization step can generate consistent appearances, the refinement stage could also introduce inconsistencies in the occluded areas, as there is no cross-frame constraint during the inpainting process.

**Questions:**

- The supplementary videos lack method names, making it difficult to identify the final result of the proposed "OccFusion" method. It is unclear why some blur exists in the visible areas, as evident in cases like "ocmotion_0011_1" and "ocmotion_0011_2".

**Limitations:**

No negative societal impact.

---

> ### Author Rebuttal · Authors · 2024-08-07
>
> We thank the reviewer for your time and the helpful comments! We address your concerns below.
>
> > Proof that applying SDS on RGB images causes appearance inconsistency
>
> In Fig. 3 of the rebuttal PDF, we include additional experiments comparing the rendering results of applying SDS on RGB vs. on human occupancy maps (as proposed). It is clear that applying SDS on RGB leads to defective renderings in the Optimization stage, which ultimately cause inferior results in the Refinement stage. We attribute these defective renderings to the variances introduced from diffusion models, making it difficult for them to reconstruct a consistent human appearance across different frames and training steps. However, applying SDS on human occupancy maps does not suffer from these variances too much and allows for the enforcing of human completeness, as proposed.
>
> > Refinement stage could introduce cross-frame inconsistency
>
> It is true that diffusion-based inpainting models suffer from inconsistencies. However, $\{\mathbf{\hat{I}}\}$, the renderings from the optimization stage, are consistent since they are recovered directly from the input video. As we use these consistent images as context for our inpainting, the resulting inpainted reference images are not dramatically inconsistent. It is also worth noting that any small inconsistencies present in the inpainted references are smoothed out by more weight on perceptual loss terms and the usage of L1 loss during optimization of the Refinement stage (lines 227-230 of the main paper).
>
> >Supplementary videos lack method names
>
> Sorry for the inconvenience. We included the naming details in section “D” of the supplementary material, but we can see how it can be confusing. We will be sure to label the videos with method names in the final version.
>
> > Blur present in visible areas
>
> With only 10 mins of training time, OccFusion surpasses state-of-the-art occluded human rendering methods by a significant margin qualitatively and quantitatively. In addition, with longer training time, OccFusion is able to render in higher quality with much less blur. We add experiments to validate this statement – please see qualitative and quantitative results in Fig. 1 in the rebuttal PDF. We believe that our proposed 10 minute version of OccFusion is able to achieve the best balance of rendering quality and efficiency.

---

> > ### Comment · Reviewer_gVDG · 2024-08-13
> >
> > Thanks for the author's rebuttal. They addressed most of my concerns.
> >
> > However, I share similar reservations with reviewer-BvJA regarding the rendering quality. As illustrated in the attached PDF file, extended optimization times reduce the blurs but introduce new artifacts, aliasing along the edges. In addition, a fitting time of 60 minutes for a GS is relatively time-consuming.
> >
> > Given the feedback from other reviewer and my ongoing concern, I will maintain my current score of 5.

---

> > > ### Author Response · Authors · 2024-08-13
> > >
> > > We thank the reviewer for your continued feedback! We are glad that we were able to resolve most of your concerns with our method.
> > >
> > > We would like to reiterate that with 10 minutes of training time, OccFusion significantly surpasses the previous state-of-the-art for occluded human rendering. In addition, for unoccluded scenes, OccFusion performs consistent with GauHuman, among the state-of-the-art in Gaussian-splatting based human rendering.
> > >
> > > We understand that 60 minutes for a GS is relatively time consuming. So, for the camera-ready version, we will focus on presenting the 10-minute version, as it offers a strong balance between efficiency and rendering quality while still outperforming existing methods.
> > >
> > > We would like to once again express our heartfelt thanks for your helpful feedback and suggestions. We would be happy to continue the discussion if the reviewer has additional questions or concerns.

---

### Author Rebuttal · Authors · 2024-08-07

We would like to thank all of the reviewers for their thoughtful feedback and helpful suggestions! We agree that occluded human reconstruction from a monocular video is an “important and unexplored problem” (Reviewer BvJA) that is “relatively new and practical” (Reviewer gVDG). By proposing a “novel”, “well motivated”, and “intuitive” (Reviewer Vqeb) three-stage architecture composed of Initialization, Optimization, and Refinement stages, our method is not only “efficient” (Reviewer WHSE), but also “demonstrates superior performance compared to previous methods” (Reviewer 9Cmr).

In the rebuttal, we clarify some concerns raised by the reviewers and, as requested, show some additional experiments to address the reviewers’ shared concerns:

> Effect of training time on rendering quality (Reviewer BvJA, gVDG, 9Cmr)

We explore the capacity of the models by training both OccFusion and OccGauHuman for 60 minutes, compared to the 10 minutes reported in the paper. After 60 minutes of training, both of the methods are fully converged. As shown in Fig. 1 in the rebuttal PDF, OccGauHuman is still unable to recover occluded regions of the human. However, training OccFusion for 60 minutes leads to less blurs in the final renderings. It also achieves slightly better quantitative performance, with a higher PSNR and lower LPIPS. Notably, with only 10 minutes of training, OccFusion is able to achieve on-par performances, yielding the best efficiency-performance trade off.

> Robustness of training to ‘wrong’ masks (Reviewer BvJA)

For in-the-wild occluded videos, there are no ‘ground truth’ masks for the occluded body regions due to unknown human/garment deformations. Relying on the state-of-the-art pre-trained priors brought by the Segment Anything model (SAM) and Stable Diffusion, the segmented/inpainted masks are expected to be reasonable and coherent across frames. To test the robustness of our method to variances in the in-painted masks, we add comparison experiments on ZJU-MoCap that supervise using the complete SAM masks obtained from the **unoccluded humans** with minimum variances.  Please see the qualitative and quantitative results in Fig. 2 in the rebuttal PDF. We find that using the inpainted $\hat{M}$ leads to a good enough rendering quality comparable to using masks derived from the unoccluded images, validating the robustness of our model.


> Applying SDS on RGB vs on Human Occupancy Maps (Reviewer gVDG)

In Fig. 3 of the rebuttal PDF, we include additional experiments comparing the rendering results of applying SDS on RGB vs. on human occupancy maps (as proposed). It is clear that applying SDS on RGB leads to defective renderings as well as inferior quantitative results. This experiment validates our claim made in Sec. 4.1 and Fig. 4 in the main paper.

> Applying SDS on Canonical and Deformed Spaces Only vs. Jointly (Reviewer Vqeb)

In the Optimization Stage, we apply Score Distillation Sampling (SDS) to both the deformed and canonical spaces (see Sec. 4.2 and Eq. 5 of the main paper). The space is controlled by a random variable with a 75% probability of applying SDS on the deformed space and 25% on the canonical space. We add additional experiments to show that our proposed random mix of deformed/canonical space SDS yields the best results compared to applying SDS solely on the deformed space or the canonical space (please see Fig. 4 in the rebuttal PDF).

---

### Decision · Program_Chairs · 2024-09-25

**Decision:**

Accept (poster)

**Comment:**

The paper received good scores. In the rebuttal phase, the authors engaged in fruitful discussions with the reviewers. Many of the smaller concerns were removed, and the reviewers agreed to recommend accepting the work. AC agrees. Congrats.

Please include the additional clarification, as brought up during the rebuttal phase, while preparing the final version of the paper.